# Dialect-to-Standard Normalization: A Large-Scale Multilingual Evaluation

**Olli Kuparinen[1,2]**
first.last@tuni.fi

**Aleksandra Miletić[1]**
first.last@helsinki.fi

**Yves Scherrer[1,3]**
first.last@ifi.uio.no

[1]Department of Digital Humanities, University of Helsinki
[2]Faculty of Information Technology and Communication Sciences, Tampere University
[3]Department of Informatics, University of Oslo

## Abstract

Text normalization methods have been commonly applied to historical language or user-generated content, but less often to dialectal transcriptions. In this paper, we introduce dialect-to-standard normalization – i.e., mapping phonetic transcriptions from different dialects to the orthographic norm of the standard variety – as a distinct sentence-level character transduction task and provide a large-scale analysis of dialect-to-standard normalization methods. To this end, we compile a multilingual dataset covering four languages: Finnish, Norwegian, Swiss German and Slovene. For the two biggest corpora, we provide three different data splits corresponding to different use cases for automatic normalization. We evaluate the most successful sequence-to-sequence model architectures proposed for text normalization tasks using different tokenization approaches and context sizes. We find that a character-level Transformer trained on sliding windows of three words works best for Finnish, Swiss German and Slovene, whereas the pre-trained byT5 model using full sentences obtains the best results for Norwegian. Finally, we perform an error analysis to evaluate the effect of different data splits on model performance.

## 1 Introduction

Text normalization refers to a range of tasks that consist in replacing non-standard spellings with their standard equivalents. This procedure is beneficial for many downstream NLP tasks, since it increases data homogeneity and thus reduces the impact of unknown word forms. Establishing identity between different variants of the same form is also crucial for information retrieval tasks, and in particular for building efficient corpus querying systems. Furthermore, it can facilitate building applications for a wider audience, such as spelling and grammar checkers.

Important progress has been made on normalization of historical texts (e.g., Tang et al., 2018; Boll-mann, 2019) and user-generated content (UGC) in social media (e.g. van der Goot et al., 2021). However, the existing work on dialect normalization (e.g., Scherrer and Ljubešić, 2016; Abe et al., 2018; Partanen et al., 2019) remains fragmented: it typically focuses on a single language and uses different models, experimental setups and evaluation metrics, making direct comparisons difficult.

In this paper, we aim to establish dialect-to-standard normalization as a distinct task alongside historical text normalization and UGC normalization. We make the following contributions:

• We compile a **multilingual dataset** from existing sources and make it available in a unified format to facilitate cross-lingual comparisons. The dataset covers Finnish, Norwegian, Swiss German and Slovene. These languages come from three different families and have different morphological systems. For the two largest datasets (Finnish and Norwegian), we provide **different data splits** corresponding to different use cases for dialect normalization.

• We test a **wide range of sequence-to-sequence models** that performed well in other normalization tasks: statistical machine translation, RNN-based and Transformer-based neural machine translation, and byT5, a pre-trained byte-based multilingual Transformer. We compare character with subword tokenizations as well as full-sentence contexts with sliding windows of three words. We evaluate the models on word accuracy, but also provide character error rates and word error reduction rates to facilitate comparison with previous work. Finally, we provide an error analysis on two of the Finnish data splits.[1]

---

[1]The data, scripts and model configurations are available at https://github.com/Helsinki-NLP/dialect-to-standard.

## 2 Related Work

### 2.1 Historical text normalization

Historical text normalization consists in modernizing the spelling of the text such that it conforms to current orthographic conventions. Pettersson et al. (2014) evaluate three different normalization methods in a multilingual setup: a simple filtering model, an approach based on Levenshtein distance, and an approach using character-level statistical machine translation (CSMT). They find that CSMT is the overall most promising approach. Scherrer and Erjavec (2016) use CSMT in supervised and unsupervised settings to normalize historical Slovene data.

Tang et al. (2018) and Bollmann (2019) provide multilingual comparisons of neural and statistical MT approaches, whereas Bawden et al. (2022) evaluate different normalization methods on historical French. In most settings, SMT outperformed neural models. In several settings, BPE-based subword segmentation led to better results than character-level segmentation.

### 2.2 Normalization of user-generated content

UGC, typically found on social media, contains various non-standard elements such as slang, abbreviations, creative spellings and typos. De Clercq et al. (2013) present various experiments on normalizing Dutch tweets and SMS messages and show that a combination of character-level and word-level SMT models yields the optimal results. Matos Veliz et al. (2019) follow up on this work and show that data augmentation techniques are crucial for obtaining competitive results with NMT models. The MoNoise model (van der Goot, 2019) significantly improved the state-of-the-art in UGC normalization. It contains several modules such as a spellchecker, an n-gram language model and domain-specific word embeddings that provide normalization candidates.

The first multilingual, homogeneous dataset for UGC normalization was published in the context of the MultiLexNorm shared task in 2021 (van der Goot et al., 2021). The results of the shared task also supported the usefulness of normalization for downstream tasks such as PoS-tagging and parsing. The best-performing submission (Samuel and Straka, 2021) proposed to fine-tune byT5, a byte-level pre-trained model (Xue et al., 2022), in such a way that normalizations are produced one word at a time.

### 2.3 Dialect-to-standard normalization

There has been comparatively less research in the domain of dialect normalization. Scherrer and Ljubešić (2016) apply CSMT to Swiss German. They create models for normalizing individual words and entire sentences and show that the larger context provided by the latter is beneficial for the normalization of ambiguous word forms. Lusetti et al. (2018) work on a different Swiss German dataset and show that neural encoder-decoder models can outperform CSMT when additional target-side language models are included.

Abe et al. (2018) work on the NINJAL corpus (Kokushokankokai Inc, 1980), and propose to translate Japanese dialects into standard Japanese using a multilingual (or rather, multi-dialectal) LSTM encoder-decoder model. The transduction is done on the level of *bunsetsu*, the base phrase in Japanese, which corresponds to a content word, potentially followed by a string of functional words.

Partanen et al. (2019) compare LSTM-based and Transformer-based character-level NMT models for normalizing Finnish. The authors use contexts of one word, three words, or the entire sentence, the best results being achieved using three words. On the contrary, Hämäläinen et al. (2020) report optimal results while using individual words in normalizing Swedish dialects spoken in Finland. Hämäläinen et al. (2022) use generated dialectal Finnish sentences to normalize Estonian dialects.

Machine translation from Arabic dialects to Modern Standard Arabic (MSA) can also be considered a dialect normalization task, although Arabic dialects differ from the standard variety to a greater extent than the languages used in our work and therefore contain lexical replacements and reorderings. The 2023 NADI shared task (Abdul-Mageed et al., 2023) provides a subtask on Arabic dialect translation. The MADAR corpus (Bouamor et al., 2018) is a popular resource for Arabic dialect translation and covers 25 dialects. Additionally, Eryani et al. (2020) describe the creation of a normalization corpus for five Arabic dialects, but do not report any experiments on automatic normalization. Zhao and Chodroff (2022) similarly report on corpus compilation of Mandarin dialects, but they focus on acoustic-phonetic analysis.

Each of these works focuses on dialects of a single language and uses different models and experimental setups, as well as different evaluation metrics (including BLEU, word error rate and ac-

|          | Speak. | Loc. | Texts | Sentences | Words     | Types   | MSL$_w$ | MSL$_{ch}$ | Ambig. |
|----------|--------|------|-------|-----------|-----------|---------|---------|------------|--------|
| SKN      | 99     | 50   | 99    | 41,407    | 630,665   | 106,452 | 15.23   | 88.59      | 5.54%  |
| NDC      | 438    | 111  | 684   | 126,460   | 1,684,059 | 77,950  | 13.32   | 57.16      | 10.14% |
| ArchiMob | 6      | 5    | 6     | 10,183    | 82,658    | 11,110  | 8.12    | 42.17      | 6.36%  |
| GOS      | 36     | 10   | 24    | 8,621     | 84,199    | 12,616  | 9.77    | 41.43      | 8.85%  |

Table 1: Key figures of the four datasets constituting the multilingual normalization benchmark. Speak. = number of speakers. Loc. = number of locations. MSL$_w$ = mean sentence length in words. MSL$_{ch}$ = mean sentence length in characters. Ambig. = percentage of types having more than one possible normalization.

| SKN (Finnish) | mä | oon | syänys | seittemän | silakkaa | aiva | niin | häntä | erellä | | |
|---|---|---|---|---|---|---|---|---|---|---|---|
| | minä | olen | syönyt | seitsemän | silakkaa | aivan | niin | häntä | edellä | | |
| | 'I have eaten seven herrings, that's right, tail first' | | | | | | | | | | |

| NDC (Norwegian) | å | får | eg | sje | sjøra | vår | bil | før | te | påske | |
|---|---|---|---|---|---|---|---|---|---|---|---|
| | og | får | jeg | ikke | kjøre | vår | bil | før | til | påske | |
| | 'and I don't get to drive our car until Easter' | | | | | | | | | | |

| ArchiMob (Swiss German) | ich | ha | das | ales | inere | kasette | won | ich | de | schlüssel | nüme | | ha | dezue |
|---|---|---|---|---|---|---|---|---|---|---|---|---|---|---|
| | ich | habe | das | alles | in einer | kasette | wo | ich | den | schlüssel | nicht mehr | habe | dazu |
| | 'I have it all in a case for which I don't have the key anymore' | | | | | | | | | | | | | |

| GOS (Slovene) | se | zjemla | je | prpravlena | pugnujena | pa | ubdajlana | pa | puvlajčena | |
|---|---|---|---|---|---|---|---|---|---|---|
| | saj | zemlja | je | pripravljena | pognojena | pa | obdelana | pa | povlečena | |
| | 'because the soil is prepared, fertilised and tilled and harrowed' | | | | | | | | | |

Table 2: Normalization examples of the four languages. The top row presents the original phonetic transcription, the middle row the normalized version, and the bottom row provides an English gloss.

curacy). All of these factors make meaningful comparisons between different approaches difficult.

## 3 Datasets

We propose a multilingual dataset that covers Finnish, Norwegian, Swiss German and Slovene. The dataset is compiled from existing dialect corpora, which are presented in detail below. The languages originate from two language families (Uralic and Indo-European) and three branches (Finnic, Germanic, Balto-Slavic). All languages adopt the Latin script. This enables the comparison of language structure, rather than differences in script. Some quantitative information about the individual corpora is available in Table 1, and Table 2 provides some example sentences.

### 3.1 Finnish

The Samples of Spoken Finnish corpus (*Suomen kielen näytteitä*, hereafter SKN) (Institute for the Languages of Finland, 2021) consists of 99 interviews conducted mostly in the 1960s.[2] It includes data from 50 Finnish-speaking locations, with two speakers per location (with one exception). The interviews have been transcribed phonetically on two levels of granularity (detailed and simplified) and normalized manually by linguists. We only consider the utterances of the interviewed dialect speakers, not of the interviewers. Although the detailed transcriptions have been used for the normalization experiments in Partanen et al. (2019), we use the simplified transcriptions here to make the annotations more consistent with the other languages. The simplified transcriptions do not make certain phonetic distinctions and share the alphabet with the normalized text.

### 3.2 Norwegian

The Norwegian Dialect Corpus (Johannessen et al., 2009, hereafter NDC) was built as a part of a larger initiative to collect dialect syntax data of the North Germanic languages.[3] The recordings were made between 2006 and 2010, and typically four speakers per location were recorded. Each speaker appears in an interview with a researcher and in an informal conversation with another speaker. We concatenate all utterances of a speaker regardless of the context in which they appear.

---

[2]http://urn.fi/urn:nbn:fi:lb-2021112221, Licence: CC-BY.

[3]http://www.tekstlab.uio.no/scandiasyn/download.html, Licence: CC BY-NC-SA 4.0.

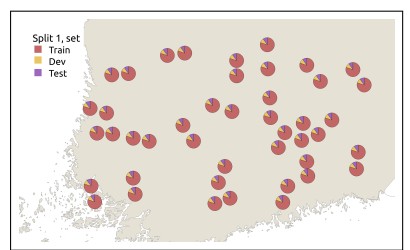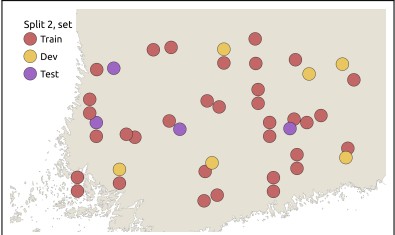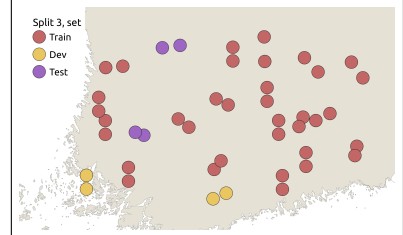

Figure 1: The three data splits (SKN1 on the left, SKN2 in the center, SKN3 on the right) visualized on a subset of the Finnish dataset. Each dot represents one speaker. There are generally two speakers per location in SKN.

The recordings were transcribed phonetically and thereafter normalized to Norwegian Bokmål. The normalization was first done with an automatic tool developed specifically for this corpus, and its output was manually corrected afterwards. The publicly available phonetic and orthographic transcriptions are not well aligned; we automatically re-aligned them at utterance and word level.[4]

### 3.3 Swiss German

The ArchiMob corpus of Swiss German (Samardžić et al., 2016; Scherrer et al., 2019) consists of oral history interviews conducted between 1999 and 2001.[5] The corpus contains 43 phonetically transcribed interviews, but only six of them were normalized manually. We use the interviewee's utterances of these six documents for our experiments. The selected texts originate from five dialect areas, covering approximately one third of the German-speaking part of Switzerland.

### 3.4 Slovene

Our Slovene dataset is based on the GOS corpus of spoken Slovene (Verdonik et al., 2013).[6] The original corpus contains 115h of recordings. Two transcription layers are included: a manually transcribed phonetic layer and a semi-automatically normalized layer (with manual validation).

Since the degree of non-standardness in the full corpus was relatively low (16%), we select a subset of the data for our experiments. We retain speakers whose productions contain at least 30% non-standard tokens and who have produced at least 1000 words. This results in a set of 36 speakers from 10 dialect regions.

[4]The alignment script and re-aligned data is available at https://github.com/Helsinki-NLP/ndc-aligned/.
[5]https://www.spur.uzh.ch/en/departments/research/textgroup/ArchiMob.html, Licence: CC BY-NC-SA 4.0.
[6]http://hdl.handle.net/11356/1438, Licence: CC BY-NC-SA 4.0.

### 3.5 Preprocessing

To ensure the datasets are comparable, we have applied several preprocessing steps: removing punctuation and pause markers, substituting anonymized name tags with X, and excluding utterances consisting only of filler words. The Slovene data includes some utterances in Italian and German which are normalized to the corresponding standard. They have been excluded from the data.

## 4 Experimental Setup

### 4.1 Data splits

The Finnish and Norwegian datasets contain multiple speakers per location, which provides a possibility to test the generalization capabilities of the models in different scenarios. We create three different data splits:

1. **Normalizing unseen sentences of seen speakers.** We divide each speaker's data in such a way that 80% of *sentences* are used for training, 10% for development and 10% for testing. The sentences are selected randomly.

2. **Normalizing unseen speakers of seen dialects.** We pick speakers from selected locations for the development and test sets, while the rest of the speakers are used for training. For each location, at least one speaker is present in the training set. In other words, 80% of *speakers* are used for training, 10% for development and 10% for testing.

3. **Normalizing unseen dialects.** All speakers from a given location are assigned to either the training, development or test set. In other words, 80% of *locations* are used for training, 10% for development and 10% for testing.

The different data splits are visualized in Figure 1. For the smaller and less geographically diverse Swiss German and Slovene datasets, we only

use split 1. For each split, we create three folds with random divisions into train, development and test sets.

While previous work (Scherrer and Ljubešić, 2016; Partanen et al., 2019) mostly relies on split 1, this setup potentially overestimates the models' normalization capabilities: in a given conversation, utterances, phrases and words are often repeated, so that similar structures can occur in the training and test sets. We argue that splits 2 and 3 more realistically reflect dialectological fieldwork, where new texts are gradually added to the collection and need to be normalized.

## 4.2 Tokenization and context sizes

Text normalization is generally viewed as a character transduction problem (Wu et al., 2021), and it seems therefore most natural to use single characters as token units. Character tokenization has also been shown to work well for normalization tasks in various recent studies.

Most other character transduction problems, such as transliteration or morphological inflection, are modelled out-of context, i.e., one word at a time. This assumption does not seem accurate for text normalization: as shown in Table 1, between 5 and 10% of word types have more than one possible normalization, and disambiguating these requires access to the sentential context. Moreover, depending on the annotation scheme, there are sandhi phenomena at the word boundaries (cf. the SKN example in Table 2: *syänys* instead of *syänyt* because of assimilation with the following *s*) that cannot be taken into account by models that operate word by word. Thus, the most obvious strategy for text normalization is to consider **entire sentences** and break them up into **single character tokens**.

This strategy combining long contexts with short tokens leads to rather long token sequences, and NMT approaches especially have been shown to underperform in such scenarios (Partanen et al., 2019). We include two alternative ways of addressing this issue: (1) by shortening the instances from full sentences to sliding windows of three consecutive words, and (2) by lengthening the tokens using subword segmentation.

**Sliding windows.** Partanen et al. (2019) propose to split each sentence into non-overlapping chunks of three consecutive words. We adapt this approach and use **overlapping chunks of three words** to ensure that the model always has access to exactly

one context word on the left and one on the right. At prediction time, only the word in the middle of each chunk is considered.

The preprocessing needed to create input for both entire-sentence and sliding-window models is illustrated in Appendix D.

**Subword segmentation.** Tang et al. (2018) and Bawden et al. (2022) found that subword segmentation could outperform character-level segmentation on the task of historical text normalization. We follow this work and experiment with subword segmentation as well. Several segmentation schemes have been proposed for general machine translation, e.g. byte-pair encoding (Sennrich et al., 2016, BPE) or the unigram model (Kudo, 2018). Kanjirangat et al. (2023) found the unigram model to perform better than BPE on texts with inconsistent writing. This is the case for our speech transcriptions and we thus opt to use the unigram model. We train our models with the SentencePiece library (Kudo and Richardson, 2018), and optimize the vocabulary size separately for each dataset.

## 4.3 Evaluation

We evaluate the models on **word-level accuracy**, i.e., the percentage of correctly normalized words. Since the reference normalizations are tied to the words in the source sentences, and since the models' output can differ in length from the source sentence, we need to re-align the model output with the reference normalization. We apply Levenshtein alignment to the entire sequence pair and split the system output at the characters aligned with word boundaries of the input (see Appendix D for an illustration).[7]

Word-level accuracy lacks granularity and does not distinguish between normalizations that are only one character off and normalizations that are completely wrong. Therefore, we include **character error rate** (CER) as a more fine-grained metric.[8] CER is defined as the Levenshtein distance between the system output and the reference, normalized by the length of the reference. Another advantage of CER is that it can be computed directly on sentence pairs without re-alignment.[9]

---

[7]The re-alignment code is provided at https://github.com/Helsinki-NLP/dialect-to-standard/blob/main/scripts/align.py.

[8]Following Partanen et al. (2019), we use the implementation available at https://github.com/nsmartinez/WERpp.

[9]Re-alignment is still necessary for the sliding window models in order to extract the center word of each chunk.

Following van der Goot et al. (2021), we also report error reduction rates in Appendix C.

We compare the systems to two **baselines**: the **leave-as-is** (LAI) baseline corresponds to the percentage of words that do not need to be modified. The **most frequent replacement** (MFR) baseline translates each word to its most frequent replacement seen in the training data, and falls back to copying the input for unseen words.

## 5 Methods and Tools

We utilize both statistical and neural machine translation tools trained from scratch, as well as a pretrained multilingual model. We train (or fine-tune) the models for each of the four languages separately. Our hyperparameter choices largely follow recent related work on text normalization. The main characteristics of the models are summarized below, and a detailed description of the hyperparameters is given in Appendix A. The methods used in the experiments are:

**SMT.** Our statistical machine translation method corresponds mostly to the one implemented in the CSMTiser tool.[10] It uses the Moses SMT toolkit (Koehn et al., 2007) with a 10-gram KenLM language model trained on the training sets.[11] Scherrer (2023) found eflomal (Östling and Tiedemann, 2016) to produce better character alignment than the more commonly used GIZA++, and we adopt this method. Minimum error rate training (MERT) is used for tuning the model weights, using WER (word error rate, which effectively becomes character error rate in a character-level model) as the objective.

**RNN-based NMT.** This model uses a bidirectional LSTM encoder and a unidirectional LSTM decoder with two hidden layers each. The attention mechanism is reused for copy attention.

**TF-based NMT.** This model has 6 Transformer layers in the encoder and the decoder, with 8 heads each. We found in preliminary experiments that position representation clipping was beneficial to the results. All NMT models are trained with the OpenNMT-py toolkit (Klein et al., 2017).

**ByT5.** ByT5 is a multilingual pre-trained model of the T5 family (Raffel et al., 2020). These models use a Transformer architecture and are pretrained on a masked language modeling task. ByT5 (Xue et al., 2022) is a variant of T5 that does not use any subword tokenization model, but rather encodes all text as UTF-8 encoded byte sequences.[12] It is pre-trained on the multilingual m4C corpus (Xue et al., 2021), which includes the four languages of our dataset. We use the `byt5-base` model and fine-tune it separately on each normalization task for ten epochs.

We run all models on our base setup (entire sentence instances with character or byte tokenization). In addition, the SMT, RNN and TF models are also trained on subword-segmented data. Finally, the character-level RNN and TF models are trained on sliding windows.[13]

## 6 Results and Discussion

We evaluate the models presented in Section 5 with the metrics described in Section 4.3. We run the models on the three folds of each data split, and present the average scores and standard deviations for each metric.

The word-level accuracies are presented in Table 3 and the character error rates in Table 4. We also provide accuracy scores for the development sets in Appendix C. Transformer-based methods appear as the most robust: the Transformer trained on sliding windows is best for SKN, ArchiMob and GOS, while the Transformer-based byT5 produces the best results for Norwegian. For all other corpora, byT5 is a close second in accuracy, and is thus the best alternative for entire sentences.

While CSMT and RNN-based methods do not yield the best score for any dataset, they still perform reasonably well. For the NMT models, using the sliding window instead of sentences enhances the results for all corpora but NDC. Regarding tokenization, the subwords improve the performance on the large corpora (SKN and NDC) but worsen it on the small corpora (Archimob and GOS). For all datasets, the best results are obtained on the character (or byte) level.

We expected rising levels of difficulty between data split 1 and 3, but neither the baselines nor the model outputs confirmed our expectations. The

---

[10] https://github.com/clarinsi/csmtiser
[11] Contrarily to related work, we did not include any additional target-side language models.

[12] In our setup, we view character-level and byte-level segmentation as largely equivalent. Our datasets have vocabulary sizes ranging between 42 and 90 and thus lie largely below the byte-level upper bound of 256.
[13] In preliminary experiments, the sliding window setup did not yield any benefits for SMT and byT5 models.

| Model | SKN1 | SKN2 | SKN3 | NDC1 | NDC2 | NDC3 | ArchiMob | GOS |
|---|---|---|---|---|---|---|---|---|
| LAI | $44.63^{\pm0.37}$ | $46.01^{\pm1.10}$ | $47.96^{\pm2.53}$ | $32.88^{\pm0.08}$ | $33.31^{\pm1.35}$ | $33.72^{\pm0.48}$ | $21.37^{\pm0.33}$ | $58.08^{\pm0.71}$ |
| MFR | $84.87^{\pm0.22}$ | $82.75^{\pm0.26}$ | $83.19^{\pm0.24}$ | $86.90^{\pm0.06}$ | $86.48^{\pm1.15}$ | $86.72^{\pm0.17}$ | $83.82^{\pm0.20}$ | $83.93^{\pm0.52}$ |
| **Full sentence models with character or byte segmentation** | | | | | | | | |
| SMT | $91.22^{\pm0.17}$ | $89.96^{\pm0.39}$ | $90.36^{\pm0.21}$ | $91.85^{\pm0.08}$ | $91.44^{\pm0.47}$ | $91.55^{\pm0.08}$ | $88.06^{\pm0.31}$ | $85.82^{\pm0.96}$ |
| RNN | $89.90^{\pm1.14}$ | $89.35^{\pm0.88}$ | $88.56^{\pm1.04}$ | $93.42^{\pm0.06}$ | $93.03^{\pm0.39}$ | $93.21^{\pm0.09}$ | $88.98^{\pm0.56}$ | $83.59^{\pm0.68}$ |
| TF | $92.90^{\pm0.35}$ | $91.38^{\pm0.86}$ | $89.75^{\pm1.73}$ | $93.99^{\pm0.09}$ | $93.68^{\pm0.47}$ | $93.41^{\pm0.67}$ | $89.14^{\pm0.38}$ | $85.65^{\pm0.61}$ |
| ByT5 | $92.95^{\pm0.11}$ | $91.77^{\pm1.11}$ | $91.90^{\pm0.77}$ | $\mathbf{94.68}^{\pm0.10}$ | $\mathbf{94.51}^{\pm0.28}$ | $\mathbf{94.54}^{\pm0.10}$ | $90.57^{\pm0.54}$ | $87.26^{\pm0.55}$ |
| **Full sentence models with subword segmentation** | | | | | | | | |
| SMT | $90.18^{\pm0.02}$ | $88.64^{\pm0.36}$ | $89.26^{\pm0.13}$ | $92.57^{\pm0.06}$ | $92.16^{\pm0.48}$ | $92.20^{\pm0.12}$ | $87.38^{\pm0.13}$ | $85.23^{\pm0.73}$ |
| RNN | $91.29^{\pm0.09}$ | $89.51^{\pm1.33}$ | $88.90^{\pm1.85}$ | $92.60^{\pm0.20}$ | $92.13^{\pm0.07}$ | $92.41^{\pm0.10}$ | $86.81^{\pm0.38}$ | $80.64^{\pm1.34}$ |
| TF | $92.23^{\pm1.80}$ | $90.84^{\pm2.10}$ | $91.21^{\pm1.59}$ | $94.20^{\pm0.01}$ | $93.85^{\pm0.37}$ | $93.96^{\pm0.12}$ | $87.63^{\pm0.37}$ | $84.72^{\pm0.83}$ |
| **Sliding window models with character segmentation** | | | | | | | | |
| RNN | $92.46^{\pm0.07}$ | $90.74^{\pm0.24}$ | $91.42^{\pm0.24}$ | $92.31^{\pm0.15}$ | $92.23^{\pm0.64}$ | $92.31^{\pm0.20}$ | $90.39^{\pm0.33}$ | $86.26^{\pm0.58}$ |
| TF | $\mathbf{93.44}^{\pm0.04}$ | $\mathbf{92.10}^{\pm0.52}$ | $\mathbf{92.38}^{\pm0.35}$ | $93.24^{\pm0.10}$ | $93.02^{\pm0.40}$ | $93.19^{\pm0.09}$ | $\mathbf{90.95}^{\pm0.21}$ | $\mathbf{87.30}^{\pm0.56}$ |

Table 3: Word-level accuracy (↑). We report averages and standard deviations over the three folds of each data split.

| Model | SKN1 | SKN2 | SKN3 | NDC1 | NDC2 | NDC3 | ArchiMob | GOS |
|---|---|---|---|---|---|---|---|---|
| LAI | $14.60^{\pm0.11}$ | $14.04^{\pm0.42}$ | $13.21^{\pm0.95}$ | $24.54^{\pm0.05}$ | $24.28^{\pm0.72}$ | $23.78^{\pm0.20}$ | $32.45^{\pm0.08}$ | $14.95^{\pm0.27}$ |
| MFR | $4.68^{\pm0.06}$ | $5.35^{\pm0.13}$ | $5.18^{\pm0.10}$ | $5.71^{\pm0.02}$ | $5.94^{\pm0.63}$ | $5.75^{\pm0.12}$ | $7.10^{\pm0.11}$ | $6.31^{\pm0.30}$ |
| **Full sentence models with character or byte segmentation** | | | | | | | | |
| SMT | $2.64^{\pm0.07}$ | $3.00^{\pm0.18}$ | $2.91^{\pm0.11}$ | $3.07^{\pm0.03}$ | $3.28^{\pm0.21}$ | $3.16^{\pm0.04}$ | $3.63^{\pm0.19}$ | $5.30^{\pm0.39}$ |
| RNN | $4.46^{\pm1.64}$ | $5.88^{\pm1.31}$ | $5.21^{\pm1.49}$ | $2.65^{\pm0.09}$ | $3.49^{\pm0.50}$ | $3.10^{\pm0.52}$ | $4.21^{\pm1.37}$ | $23.20^{\pm10.72}$ |
| TF | $2.39^{\pm0.20}$ | $3.13^{\pm0.67}$ | $4.57^{\pm1.38}$ | $2.32^{\pm0.03}$ | $2.48^{\pm0.22}$ | $2.59^{\pm0.33}$ | $3.49^{\pm0.27}$ | $5.71^{\pm0.21}$ |
| ByT5 | $2.96^{\pm0.13}$ | $3.67^{\pm1.12}$ | $3.75^{\pm0.99}$ | $\mathbf{2.11}^{\pm0.06}$ | $\mathbf{2.19}^{\pm0.12}$ | $\mathbf{2.19}^{\pm0.08}$ | $3.21^{\pm0.36}$ | $5.28^{\pm0.19}$ |
| **Full sentence models with subword segmentation** | | | | | | | | |
| SMT | $2.97^{\pm0.03}$ | $3.46^{\pm0.19}$ | $3.30^{\pm0.12}$ | $2.96^{\pm0.04}$ | $3.20^{\pm0.25}$ | $3.14^{\pm0.02}$ | $4.20^{\pm0.08}$ | $5.92^{\pm0.35}$ |
| RNN | $7.38^{\pm0.44}$ | $9.57^{\pm5.20}$ | $12.38^{\pm4.46}$ | $3.34^{\pm0.65}$ | $3.20^{\pm0.17}$ | $3.14^{\pm0.03}$ | $4.84^{\pm0.34}$ | $28.66^{\pm1.06}$ |
| TF | $2.23^{\pm0.10}$ | $2.82^{\pm0.18}$ | $2.93^{\pm0.63}$ | $2.27^{\pm0.02}$ | $2.44^{\pm0.18}$ | $2.37^{\pm0.07}$ | $4.64^{\pm0.15}$ | $7.05^{\pm0.10}$ |
| **Sliding window models with character segmentation** | | | | | | | | |
| RNN | $2.38^{\pm0.01}$ | $2.90^{\pm0.11}$ | $2.73^{\pm0.10}$ | $3.06^{\pm0.07}$ | $3.12^{\pm0.34}$ | $3.02^{\pm0.06}$ | $3.22^{\pm0.09}$ | $5.38^{\pm0.23}$ |
| TF | $\mathbf{2.06}^{\pm0.02}$ | $\mathbf{2.73}^{\pm0.64}$ | $\mathbf{2.40}^{\pm0.13}$ | $2.68^{\pm0.06}$ | $2.79^{\pm0.20}$ | $2.68^{\pm0.01}$ | $\mathbf{3.02}^{\pm0.08}$ | $\mathbf{4.98}^{\pm0.18}$ |

Table 4: Character error rates (↓).

differences between splits are very small, and most models achieve the lowest results on split 2.

The character error rates presented in Table 4 follow the same pattern as the word accuracies when it comes to the best models. However, for SKN, byT5 drops below sliding window RNN and SMT with this metric. Some of the Finnish byT5 models generate much shorter predictions than the other models, but it remains to be investigated why this occurs and why it only affects some training runs. Table 4 also highlights poor performance and large standard deviation with sentence-level RNN on GOS. This is in line with earlier findings

about neural models' tendency to overmodify the predictions (Bawden et al., 2022).

## 6.1 Comparison with previous work

Partanen et al. (2019) worked on the normalization of the Finnish SKN dataset, reporting word error rate (WER) as their main metric. Although they use the detailed SKN transcriptions instead of the simplified ones, their results are roughly comparable with our SKN1 data split (see Table 5). While they were not able to successfully train sentence-level models, our parameterization closes the gap to the chunk models. Their best reported word error rates

| | Partanen et al. (2019) | This work |
|---|---|---|
| RNN (full sentences) | 46.52 | 11.11 |
| RNN (3-word chunks) | 5.73 | 7.59 |
| TF (full sentences) | 53.23 | 7.07 |
| TF (3-word chunks) | 6.10 | 6.59 |

Table 5: Comparison with previous work on Finnish (SKN1 data split, all models with character tokenization). Metric: word error rate (↓).

are however slightly lower than ours.

Scherrer and Ljubešić (2016) presented normalization experiments on the ArchiMob corpus. Our results are largely comparable to theirs. A detailed comparison is provided in Appendix B.

## 6.2 Error analysis

We examine the effects of different data splits by looking at the output of the sliding window Transformer on splits 1 and 3 of the Finnish SKN corpus. As a reminder, the test set in split 1 contains unseen sentences from seen texts (and therefore seen dialects), whereas in split 3 it comes from unseen locations. It can be expected (1) that the model trained on SKN1 performs better on dialect-specific phenomena, such as normalizations involving diphthongs, consonant grade and inflection marks; and (2) that the two models behave similarly on phenomena that are not dialect-specific, such as capitalization and proper names.

| Error type | SKN1 | | SKN3 | |
|---|---|---|---|---|
| Capitalization | 2 | 1.1% | 2 | 0.6% |
| Character | 13 | 6.8% | 31 | 9.8% |
| Consonant grade | 3 | 1.6% | 15 | 4.7% |
| Diphthong | 9 | 2.7% | 4 | 1.3% |
| Indiscernible | 17 | 8.9% | 32 | 10.1% |
| Inflection | 88 | 46.3% | 127 | 39.9% |
| Proper name | 10 | 5.3% | 16 | 5.0% |
| Wrong target | 48 | 25.3% | 91 | 28.6% |
| Total | 190 | 100% | 318 | 100% |

Table 6: Comparison of error types between the models trained on SKN1 and SKN3.

We analyze model output on the sentences that appear in both test sets and focus on words for which at least one model produced an erroneous normalization. We identify 382 such cases. On this set of words, the SKN3 model produces a much higher number of errors (318) than the SKN1

model (190). The higher number of errors on SKN3 is coherent with the intuition that this split is more difficult, but the nature of the errors produced by the two models does not fully conform to the expectations (see Table 6). In absolute terms, the model trained on SKN3 does produce more inflection and consonant grade errors, but fewer diphthong errors. In relative terms, the SKN3 model produces a lower percentage of inflection errors than its counterpart (40% vs 46%). This seems to indicate that split 3 does not preclude the model from learning dialect-related patterns. We hypothesize that this is due to the fact that the training set contains material from the same dialect area as the test set (although not from the exact same location).

To identify the critical point beyond which the cross-lectal model performance would be clearly affected, it could be useful to introduce a fourth data split which would exclude larger dialect areas from the train set and test on unseen dialects.

## 6.3 Comparison with other tasks

| Task | LAI | ERR |
|---|---|---|
| Historical norm. | 17.5 – 85.4 | 72.1 – 89.9 |
| UGC norm. | 63.0 – 93.1 | 47.5 – 80.1 |
| Dialect norm. | 21.4 – 58.1 | 69.7 – 92.1 |

Table 7: Word-level accuracy ranges of the leave-as-is baselines (LAI) and word-level error reduction rate ranges (ERR) for historical normalization (Bollmann, 2019), UGC normalization (van der Goot et al., 2021) and dialect normalization (this work, cf. Appendix C).

As mentioned above, dialect-to-standard normalization shares fundamental properties with historical text normalization and UGC normalization. Here, we compare the respective difficulties of these three tasks. Table 7 reports the LAI ranges and ERR ranges of the best systems reported in Bollmann (2019) and van der Goot et al. (2021).

It can be seen that dialect normalization has the lowest LAI rates on average and thus requires the most changes of the three tasks. The models perform roughly equally well on both historical and dialectal normalization, whereas UGC normalization seems to be a more difficult task.

## 7 Conclusions

In this paper, we present the dialect-to-standard normalization task as a distinct task alongside historical text normalization and UGC normalization. We

introduce a dialect normalization dataset containing four languages from three different language branches, and use it to evaluate various statistical, neural and pre-trained neural sequence-to-sequence models.

In our base setup with models trained on entire sentences with character (or byte) segmentation, the pre-trained byT5 model performs best for all languages and data splits. Moving from character segmentation to subword segmentation increases the accuracies for the large datasets (SKN and NDC), but not enough to surpass byT5. In contrast, the sliding window approach outperforms byT5 on all languages except Norwegian. The superior performance of byT5 on Norwegian cannot be directly explained by the amount of training data,[14] but it is likely that the closely related languages Swedish and Danish enhance its performance. A further analysis on character error rate shows that the neural models sometimes offer very poor predictions, which are not visible when using accuracy as a metric.

In this work, we have evaluated the most common and most popular model architectures, but it would be interesting to test specific model architectures for character transduction tasks, e.g. models that put some monotonicity constraint on the attention mechanism (Wu and Cotterell, 2019; Rios et al., 2021). We defer this to future work.

Another point to be investigated in future work is data efficiency. Our training sets are relatively large in comparison with other character transduction tasks, and it would be useful to see how much the data requirements can be reduced without significantly affecting the normalization accuracy.

## Limitations

We see the following limitations of our work:

• The proposed multilingual dataset is biased towards European languages and European dialectal practices. It may therefore not generalize well to the types of dialectal variation present in other parts of the world and to transcriptions in non-Latin scripts. In particular, there is an extensive amount of research on the normalization of Arabic and Japanese dialects (e.g., Abe et al., 2018; Eryani et al., 2020). We address some of these issues in Scherrer et al. (2023).

• We voluntarily restrict our dataset to "clean" corpora, i.e., interviews transcribed and normalized by trained experts. This contrasts with other data collections specifically aimed at extracting dialectal content from social media (e.g., Ueberwasser and Stark, 2017; Mubarak, 2018; Barnes et al., 2021; Kuparinen, 2023). Such datasets compound the features and challenges of both dialect-to-standard normalization and UGC normalization.

• We did not perform extensive hyperparameter tuning in our experiments, but rather use settings that have performed well in other normalization tasks. It is therefore conceivable that the performance of NMT models in particular could be improved. Furthermore, specific model architectures for character transduction tasks have been proposed, e.g. constraining the attention to be monotonic (Wu and Cotterell, 2019; Rios et al., 2021). We did not include such architectures in our experiments since they generally only showed marginal improvements.

## Ethics statement

All our experiments are based on publicly available datasets that were costly to produce. It is important to ensure that these are appropriately acknowledged. Anybody wishing to use our dataset will also need to cite the publications of the original datasets. Details are given on the resource download page: `https://github.com/Helsinki-NLP/dialect-to-standard`.

The datasets have been anonymized where necessary. Text normalization is explicitly mentioned as a possible research task in the literature presenting the ArchiMob corpus (Samardžić et al., 2016; Scherrer and Ljubešić, 2016), and the SKN dataset has been previously used to evaluate normalization models. We are not aware of any malicious or harmful uses of the proposed dialect-to-standard normalization models.

## Acknowledgements

This work has been supported by the Academy of Finland through project No. 342859 "CorCoDial – Corpus-based computational dialectology".

The authors wish to acknowledge CSC – IT Center for Science, Finland, for generous computational resources.

---

[14]German accounts for 3.05% of the byT5 training data, Finnish for 1.35%, Norwegian for 1.33% and Slovene for 0.95%.

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

## A  Experimental details

We trained all neural models on a single NVIDIA V100 GPU. The SMT models were trained on a Xeon Gold 6230 CPU. Table 8 presents the average training time and number of parameters for a single fold of the largest of our datasets, the Norwegian NDC.

| Model | Runtime (hh:mm) | Parameters |
|---|---|---|
| SMT | 28:16 | — |
| RNN (full sentences) | 12:12 | 9.6 M |
| RNN (sliding window) | 42:28 | 9.6 M |
| TF (full sentences) | 17:27 | 25.4 M |
| TF (sliding window) | 30:19 | 25.4 M |
| ByT5 | 27:27 | 581 M |

Table 8: Training runtime (average) and number of parameters for a single character-level NDC model.

Additional details about the model architectures and hyperparameter settings are provided in Table 12.

## B  Comparison with previous work on Swiss German

Scherrer and Ljubešić (2016) presented normalization experiments with CSMT models on the Archi-Mob corpus, albeit with a different data split than in our present work. The most comparable model, a CSMT model trained on entire sentences, obtains an almost identical accuracy score compared to our results (see Table 9). Their best model, a CSMT model with an additional language model and constraints, achieves performance on par with our best model, the Transformer trained on sliding windows.

| | Scherrer and Ljubešić (2016) | This work |
|---|---|---|
| CSMT 1LM | 87.59 | 88.06 |
| CSMT 2LM+constraints | 90.46 | — |
| TF (sliding window) | — | 90.95 |

Table 9: Comparison with previous work on Swiss German. Metric: accuracy (↑). 1LM = one language model, 2LM = two language models.

## C  Additional results

Table 10 shows the word-level error reduction rates (ERR). ERR was introduced by van der Goot

| Model | SKN1 | SKN2 | SKN3 | NDC1 | NDC2 | NDC3 | ArchiMob | GOS |
|---|---|---|---|---|---|---|---|---|
| MFR | $72.67^{\pm0.27}$ | $68.04^{\pm0.99}$ | $67.63^{\pm1.96}$ | $80.48^{\pm0.09}$ | $79.74^{\pm1.33}$ | $79.97^{\pm0.22}$ | $79.40^{\pm0.17}$ | $61.67^{\pm1.08}$ |
| **Full sentence models with character or byte segmentation** | | | | | | | | |
| SMT | $84.14^{\pm0.21}$ | $81.42^{\pm0.37}$ | $81.43^{\pm1.05}$ | $87.87^{\pm0.11}$ | $87.17^{\pm0.49}$ | $87.25^{\pm0.05}$ | $84.79^{\pm0.44}$ | $66.20^{\pm1.82}$ |
| RNN | $81.75^{\pm2.14}$ | $80.29^{\pm1.27}$ | $78.01^{\pm1.98}$ | $90.20^{\pm0.10}$ | $89.56^{\pm0.38}$ | $89.76^{\pm0.10}$ | $85.97^{\pm0.80}$ | $60.84^{\pm2.06}$ |
| TF | $87.19^{\pm0.57}$ | $84.05^{\pm1.30}$ | $80.34^{\pm2.75}$ | $91.04^{\pm0.12}$ | $90.53^{\pm0.51}$ | $90.05^{\pm0.96}$ | $86.17^{\pm0.56}$ | $65.77^{\pm1.28}$ |
| ByT5 | $87.26^{\pm0.12}$ | $84.74^{\pm2.33}$ | $84.38^{\pm2.21}$ | $\mathbf{92.07^{\pm0.14}}$ | $\mathbf{91.78^{\pm0.30}}$ | $\mathbf{91.76^{\pm0.17}}$ | $87.99^{\pm0.76}$ | $69.60^{\pm1.34}$ |
| **Full sentence models with subword segmentation** | | | | | | | | |
| SMT | $82.27^{\pm0.08}$ | $78.96^{\pm0.28}$ | $79.33^{\pm0.94}$ | $88.93^{\pm0.09}$ | $87.97^{\pm0.16}$ | $88.29^{\pm0.11}$ | $77.53^{\pm0.25}$ | $64.77^{\pm1.56}$ |
| RNN | $84.27^{\pm0.18}$ | $80.59^{\pm2.11}$ | $78.66^{\pm3.58}$ | $88.98^{\pm0.31}$ | $88.32^{\pm0.06}$ | $88.51^{\pm0.22}$ | $83.21^{\pm0.47}$ | $53.79^{\pm3.78}$ |
| TF | $87.79^{\pm0.31}$ | $84.94^{\pm0.44}$ | $84.14^{\pm1.27}$ | $91.35^{\pm0.02}$ | $90.57^{\pm0.06}$ | $90.89^{\pm0.30}$ | $84.25^{\pm0.38}$ | $63.54^{\pm2.23}$ |
| **Sliding window models with character segmentation** | | | | | | | | |
| RNN | $86.37^{\pm0.08}$ | $82.83^{\pm0.51}$ | $83.48^{\pm0.88}$ | $88.54^{\pm0.23}$ | $88.36^{\pm0.72}$ | $88.41^{\pm0.22}$ | $87.76^{\pm0.45}$ | $67.22^{\pm1.07}$ |
| TF | $\mathbf{88.16^{\pm0.09}}$ | $\mathbf{85.37^{\pm0.67}}$ | $\mathbf{85.31^{\pm1.31}}$ | $89.93^{\pm0.15}$ | $89.54^{\pm0.40}$ | $89.73^{\pm0.13}$ | $\mathbf{88.48^{\pm0.33}}$ | $\mathbf{69.72^{\pm0.95}}$ |

Table 10: Error reduction rates (↑) relative to the LAI baseline.

| Model | SKN1 | SKN2 | SKN3 | NDC1 | NDC2 | NDC3 | ArchiMob | GOS |
|---|---|---|---|---|---|---|---|---|
| LAI | $44.74^{\pm0.33}$ | $46.90^{\pm1.56}$ | $45.13^{\pm2.77}$ | $32.61^{\pm0.06}$ | $32.79^{\pm0.52}$ | $32.70^{\pm0.88}$ | $21.38^{\pm0.34}$ | $57.29^{\pm0.48}$ |
| MFR | $84.94^{\pm0.20}$ | $83.42^{\pm1.14}$ | $81.88^{\pm2.53}$ | $86.84^{\pm0.09}$ | $86.45^{\pm0.17}$ | $86.14^{\pm0.17}$ | $83.59^{\pm0.15}$ | $83.92^{\pm0.38}$ |
| **Full sentence models with character or byte segmentation** | | | | | | | | |
| SMT | $91.83^{\pm0.79}$ | $90.55^{\pm0.25}$ | $89.87^{\pm1.98}$ | $91.75^{\pm0.40}$ | $91.64^{\pm0.45}$ | $90.92^{\pm0.21}$ | $88.43^{\pm0.19}$ | $86.03^{\pm0.15}$ |
| RNN | $90.09^{\pm1.58}$ | $90.02^{\pm1.06}$ | $89.18^{\pm2.15}$ | $93.45^{\pm0.15}$ | $93.20^{\pm0.38}$ | $92.62^{\pm0.14}$ | $89.33^{\pm0.27}$ | $82.63^{\pm0.86}$ |
| TF | $92.89^{\pm0.38}$ | $92.07^{\pm0.58}$ | $90.12^{\pm3.09}$ | $94.04^{\pm0.07}$ | $93.80^{\pm0.42}$ | $92.76^{\pm0.36}$ | $89.46^{\pm0.30}$ | $85.58^{\pm0.10}$ |
| ByT5 | $93.31^{\pm0.13}$ | $92.34^{\pm1.45}$ | $91.72^{\pm1.70}$ | $\mathbf{94.64^{\pm0.12}}$ | $\mathbf{94.58^{\pm0.33}}$ | $\mathbf{93.85^{\pm0.16}}$ | $90.55^{\pm0.51}$ | $86.95^{\pm0.87}$ |
| **Full sentence models with subword segmentation** | | | | | | | | |
| SMT | $90.34^{\pm0.16}$ | $89.39^{\pm0.40}$ | $88.53^{\pm2.17}$ | $92.62^{\pm0.13}$ | $92.20^{\pm0.47}$ | $91.52^{\pm0.19}$ | $87.62^{\pm0.31}$ | $85.35^{\pm0.40}$ |
| RNN | $91.51^{\pm0.18}$ | $90.20^{\pm0.99}$ | $89.52^{\pm2.15}$ | $92.60^{\pm0.30}$ | $92.34^{\pm0.28}$ | $91.67^{\pm0.25}$ | $87.22^{\pm0.34}$ | $80.48^{\pm1.00}$ |
| TF | $93.51^{\pm0.08}$ | $92.55^{\pm0.41}$ | $91.76^{\pm1.89}$ | $94.24^{\pm0.08}$ | $93.96^{\pm0.36}$ | $93.29^{\pm0.14}$ | $88.13^{\pm0.65}$ | $84.49^{\pm0.44}$ |
| **Sliding window models with character segmentation** | | | | | | | | |
| RNN | $92.58^{\pm0.12}$ | $91.39^{\pm0.35}$ | $90.71^{\pm2.18}$ | $92.32^{\pm0.05}$ | $92.25^{\pm0.46}$ | $91.64^{\pm0.23}$ | $90.28^{\pm0.06}$ | $86.03^{\pm0.24}$ |
| TF | $\mathbf{93.56^{\pm0.04}}$ | $\mathbf{92.79^{\pm0.11}}$ | $\mathbf{91.79^{\pm1.93}}$ | $93.23^{\pm0.17}$ | $93.07^{\pm0.38}$ | $92.53^{\pm0.23}$ | $\mathbf{90.91^{\pm0.16}}$ | $\mathbf{87.33^{\pm0.26}}$ |

Table 11: Accuracies obtained on the development sets (↑).

(2019) and it represents, roughly speaking, the improvement of a model relative to the LAI baseline. Thus, it makes it easier to compare models across datasets, which may not be the case with accuracy due to different LAI values. ERR is defined as follows:

$$ERR = \frac{Accuracy_{system} - Accuracy_{baseline}}{1.0 - Accuracy_{baseline}}$$

By and large, these results follow the same pattern as the accuracy and CER scores reported in Section 6.

Table 11 shows the accuracies on the development sets. They are comparable with the test set accuracies.

| Model | Parameter | Selected values | Considered alternatives |
|---|---|---|---|
| Subwords | Methodology | Unigram (SentencePiece) | BPE |
| | Vocabulary size | 200 (SKN) | 200, 500, 1000, 2000, |
| | | 500 (Archimob, GOS) | 4000 |
| | | 2000 (NDC) | |
| SMT | Alignment tool | Eflomal | GIZA++ |
| | Alignment symmetrization | grow-diag-final-and | — |
| | Language model n-gram size | 10 | — |
| | Maximum phrase length | 10 | — |
| | Distortion | disabled | — |
| | Tuning method | MERT | — |
| RNN | Encoder + decoder layers | 2 + 2 | — |
| | Encoder + decoder types | Bi-LSTM + LSTM | — |
| | Attention type | MLP with reused copy attention | — |
| | Embedding dimensions | 512 | — |
| | Hidden layer dimensions | 512 | — |
| | Dropout | 0.1 | — |
| | Optimizer | Adagrad | — |
| | Batch size / accumulate gradient | 1 * 10000 tokens | 1 * 5000, 1 * 25000 |
| | Learning rate | 0.5 | 0.01, 0.1, 0.2 |
| | Max. training sequence length | 1000 | — |
| | Max. prediction sequence length | 1000 | — |
| | Early stopping | 10 * 500 steps | — |
| | Early stopping criterion | validation accuracy | — |
| | Maximum training time | 50000 steps | — |
| TF | Encoder + decoder layers | 6 + 6 | — |
| | Attention heads | 8 | — |
| | Embedding dimensions | 512 | — |
| | Hidden layer dimensions | 512 | — |
| | Position representation clipping | 4 | no clipping |
| | Dropout | 0.1 | — |
| | Label smoothing | 0.1 | — |
| | Optimizer | Adam | — |
| | Adam $\beta 2$ | 0.998 | — |
| | Batch size / accumulate gradient | 8 * 5000 tokens | 8 * 1000, 8 * 10000 |
| | Batch normalization | tokens | — |
| | Initial learning rate | 4 | 0.1, 1.0, 2.0 |
| | Decay | Noam, 8000 warmup steps | — |
| | Max. training sequence length | 1000 | — |
| | Max. prediction sequence length | 1000 | — |
| | Early stopping | 10 * 500 steps | — |
| | Early stopping criterion | validation accuracy | — |
| | Maximum training time | 50000 steps | — |
| ByT5 | Foundation model | google/byt5-base | google/byt5-small |
| | Max. training sequence length | 512 (SKN), 256 (others) | — |
| | Max. prediction sequence length | 1024 | — |
| | Batch size | 4 (SKN), 8 (others) sentences | — |
| | Early stopping | disabled | — |
| | Training time | 10 epochs | — |
| | Model selection criterion | validation loss | — |

Table 12: Hyperparameter settings.

# D Data pre- and post-processing

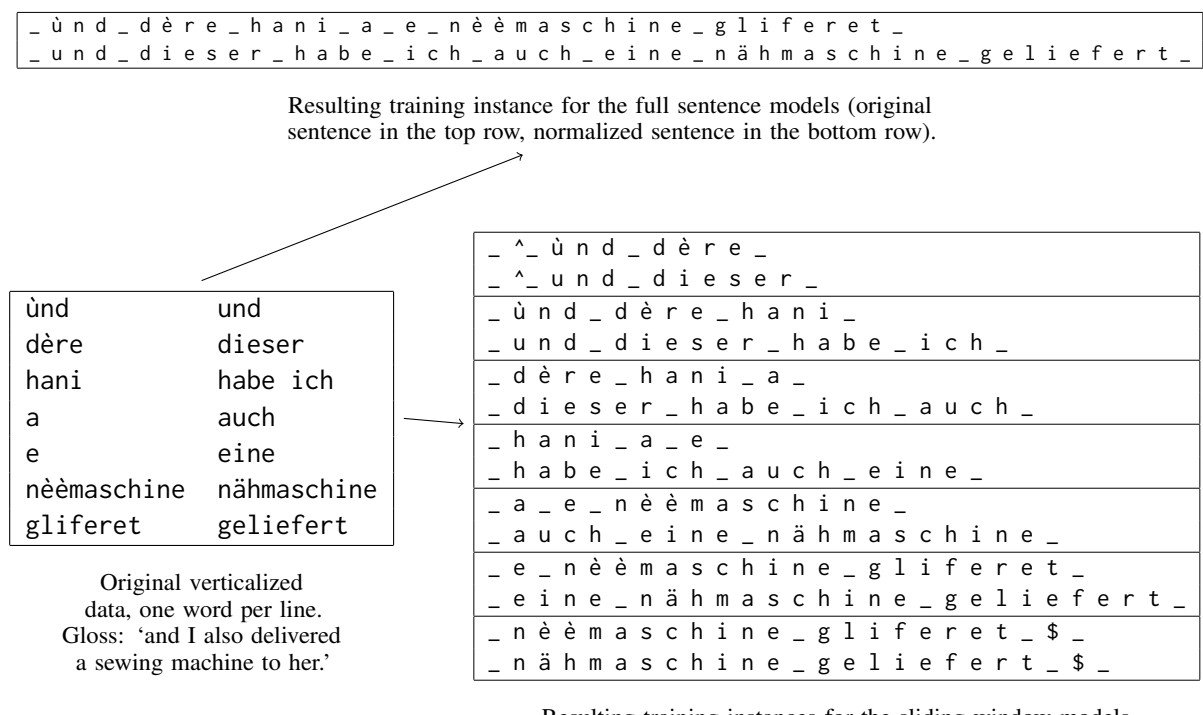

Figure 2: Data preprocessing for full sentence and sliding window models, illustrated on an example of the Swiss German ArchiMob corpus.

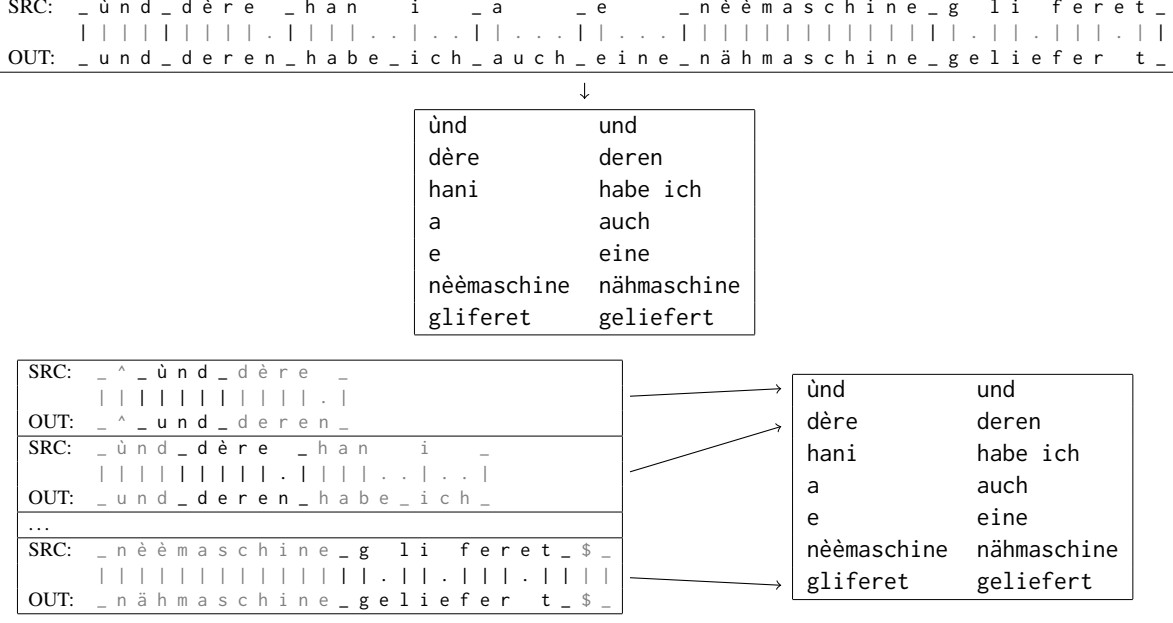

Figure 3: Realignment of normalization output (full sentence model in top half, sliding window model in bottom half). The SRC row represents the source text, the OUT row represents the system output (with erroneous *deren* instead of *dieser*). The two strings are character-aligned using Levenshtein distance, and the alignment links of the SRC word boundaries are used to separate the output words, leading to the verticalized representations.