# OpenReview forum: "Dialect-to-Standard Normalization: A Large-Scale Multilingual Evaluation"
_EMNLP/2023/Conference — EMNLP 2023 Findings_

### Official Review · Reviewer_xDQG · 2023-08-02

**Soundness:** 4

**Excitement:**

4: Strong: This paper deepens the understanding of some phenomenon or lowers the barriers to an existing research direction.

**Paper Topic And Main Contributions:**

This paper presents a benchmark for evaluating multilingual (four European languages from different families) dialect-to-standard normalization. The main task is to convert a broad phonetic transcription of dialect(s) into orthography of the standard variety. The authors evaluated several seq-2seq and MT systems and different splits and normalizations on existing datasets.


**Reasons To Accept:**

This is a well-written paper and was easy to follow and understand. There are several strengths of this work:

- Providing different splits of the same dataset to test different scenarios. This is usually rare in many NLP papers but is very important and usually highlights any artificial gains/losses in systems that can go unseen. Even though in this specific work, the splits did not affect the performance of the system, it is still helpful for future system evaluations.
- Appropriateness of the baselines and the different configurations for the systems used for this task.
- The multilingualism of the benchmark.



**Reasons To Reject:**

I have reviewed an earlier version of this paper and all the concerns that I had were addressed. No major reasons to reject this work.

**Reproducibility:**

5: Could easily reproduce the results.

**Reviewer Confidence:**

3: Pretty sure, but there's a chance I missed something. Although I have a good feel for this area in general, I did not carefully check the paper's details, e.g., the math, experimental design, or novelty.

---

> ### Author Rebuttal · Authors · 2023-08-28
>
> We thank you for recognizing positive aspects of our work and for acknowledging the evolution of the paper.

---

### Official Review · Reviewer_cje3 · 2023-08-03

**Soundness:** 2

**Excitement:**

2: Mediocre: This paper makes marginal contributions (vs non-contemporaneous work), so I would rather not see it in the conference.

**Missing References:**

[1] Mika Hämäläinen, Khalid Alnajjar, and Tuuli Tuisk. 2022. Help from the Neighbors: Estonian Dialect Normalization Using a Finnish Dialect Generator. In Proceedings of the Third Workshop on Deep Learning for Low-Resource Natural Language Processing, pages 61–66, Hybrid. Association for Computational Linguistics.

**Paper Topic And Main Contributions:**

The paper explores a large-scale multilingual study of dialect-to-standard text normalization, covering four languages. The authors compile a multilingual dataset from previous available corpora and conduct detailed experiments. With consistent settings across languages, the authors benchmark popular baselines in a unified manner and derive meaningful findings.

**Questions For The Authors:**

Question A: Do the authors conduct multilingual training/fine-tuning experiments ? Can monolingual performance be improved with resources from closely-related variants ?

Question B: The study focuses on clean transcripts (i.e. utterances normalized by trained experts). How would it generalize to informal scenarios ?

**Reasons To Accept:**

The paper compiles a multilingual dataset, and conducts consistent benchmarking across modeling methods, with detailed error analyses. The obtained results are comparable with previous works, while being more unified and observable across languages.

**Reasons To Reject:**

The paper introduces a multilingual dataset, compiled entirely of previous works on dialectal research. Regarding this aspect, the paper makes marginal contributions, as the employed corpora are derived completely from previous works, albeit with a few modifications in splitting and preprocessing.

In addition, given the availability of dialectal corpora in other languages, especially those with non-latin scripts [2][3], the compiled multilingual dataset is limited in linguistic diversity, as it’s composed from four European languages with latin scripts only, and thus the obtained findings can be biases towards these languages and do not necessarily generalize to others. The study depicts **large-scale evaluation** mostly in terms of **sample sizes**, rather than **broad linguistic coverage**.

Although the study denotes multilingual evaluation, and the languages being used are related (most belong to the Indo-European family), the experiments focus on monolingual settings rather than multilingual ones (i.e. the study benchmarks models across languages, but put little effort in cross-language linking). There were no discussions nor experiments discussing the inter-relation between languages and how multilingual training/fine-tuning, or knowledge transfer methods,  can affect monolingual performance on the task, especially between closely-related languages. In contrast, previous research has reported complementary improvement, albeit in a small-scale setting [1]. As the paper centers on multilingual evaluation, these aspects should have received more attention.



[1] Mika Hämäläinen, Khalid Alnajjar, and Tuuli Tuisk. 2022. Help from the Neighbors: Estonian Dialect Normalization Using a Finnish Dialect Generator. In Proceedings of the Third Workshop on Deep Learning for Low-Resource Natural Language Processing, pages 61–66, Hybrid. Association for Computational Linguistics.

[2] Kaori Abe, Yuichiroh Matsubayashi, Naoaki Okazaki, and Kentaro Inui. 2018. Multi-dialect Neural Machine Translation and Dialectometry. In Proceedings of the 32nd Pacific Asia Conference on Language, Information and Computation, Hong Kong. Association for Computational Linguistics.

[3] Fadhl Eryani, Nizar Habash, Houda Bouamor, and Salam Khalifa. 2020. A Spelling Correction Corpus for Multiple Arabic Dialects. In Proceedings of the Twelfth Language Resources and Evaluation Conference, pages 4130–4138, Marseille, France. European Language Resources Association.

**Reproducibility:**

4: Could mostly reproduce the results, but there may be some variation because of sample variance or minor variations in their interpretation of the protocol or method.

**Reviewer Confidence:**

4: Quite sure. I tried to check the important points carefully. It's unlikely, though conceivable, that I missed something that should affect my ratings.

---

> ### Author Rebuttal · Authors · 2023-08-28
>
> __*Lack of linguistic diversity in the sample and unclear generalization capacities of our models*__
>
> You state that our sample lacks linguistic diversity since it only uses data in Latin script in four languages “from the European family” which you consider “highly related”. We contest this characterization. Finnish is typologically and phylogenetically distinct from the other three languages. It belongs to the Uralic language family, and not the Indo-European family. It is also an agglutinative language, while the other three are fusional. And while Norwegian and Swiss German are both from the Germanic branch of the Indo-European family, Slovene belongs to the Balto-Slavic branch, known to have a richer inflectional morphology. These differences are highly relevant for the normalization task (and NLP in general) because these morphological systems tend to yield different amounts of unique surface forms. We recognize that this aspect of the dataset is not sufficiently described in the current version of the paper. We will amend this in the final version.
>
> When it comes to working only on Latin-based scripts, we are well aware of this aspect of our work and address it in the Limitations section (l. 573). We do not make claims about the generalizability of our methods to other writing systems. Indeed, it can be reasonably expected that non-alphabetic writing systems would require different techniques. We therefore preferred to focus on an in-depth evaluation and analysis of the four languages in our dataset. Introducing other scripts and languages is a promising avenue for future work but is out of the scope of this paper.
>
> Finally, note that the data used in Abe et al. (2018) is available under restrictive licensing conditions that would have precluded us from remixing and redistributing the dataset.
>
>
>
> __*Not exploring multilingual learning approaches*__
> __*Question A: Do the authors conduct multilingual training/fine-tuning experiments ? Can monolingual performance be improved with resources from closely-related variants ?*__
>
> We respectfully disagree with your statement that we do not explore multilingual settings. In fact, one of the four models we use is byT5, which is a multilingual language model. It is actually our best performing model across the board. It is only surpassed by other techniques when sliding trigram windows are used as context, and even in this setup it scores first on Norwegian. We explicitly discuss the presence of related languages in byT5’s training data as the reason for its performance (l.541-553).
>
> Given the linguistic differences between these four languages that we cited above, we do not expect significant effects through cross-lingual transfer learning based on them. Furthermore, the languages we work with can be considered as high resource when it comes to dialectal data. This is why Hämäläinen et al. (2018) use Finnish to boost performances on Estonian. We believe it is unlikely that working in the opposite direction would yield significant improvements.
>
> __*Marginal contributions when it comes to the dataset*__
>
> We explicitly state in the Introduction (l.055) that the dataset is built from existing resources. Our goal was not to produce a novel resource but rather to unify the existing ones and make them available in a format that will facilitate further multilingual research and cross-lingual comparisons for this task. We consider our main contributions to be the definition of the evaluation protocol and the tested models. We will clarify this in the final version of the paper.
>
> __*Question B*__: We have also conducted experiments on dialectal data collected from social networks in one of the languages used in the present study. The best overall results were achieved by byT5 and SMT, with Transformer- and RNN-based models scoring significantly lower. For some dialects, there were also significant differences between the top two methods. We will add the reference to the final version of the paper.
>
> __*Missing References*__: We kindly thank you for pointing out that this work is not referenced in our paper. We will add it to the final version.
>
> To conclude, we agree that some aspects of our work are suboptimal, but we challenge their status as reasons to reject. We believe that these limitations are not fundamental flaws in our research design that affect the soundness of our work. Since you do not cite any major concerns with our experimental setup or the interpretation of the results, we find it difficult to understand the soundness grade of 2 and the reproducibility grade of 3. We would appreciate any clarifications on these points to help us improve the paper.

---

### Official Review · Reviewer_fuGZ · 2023-08-05

**Soundness:** 4

**Excitement:**

4: Strong: This paper deepens the understanding of some phenomenon or lowers the barriers to an existing research direction.

**Paper Topic And Main Contributions:**

The paper discusses normalizing dialectal text to standard orthography. It presents a method based on sequence-to-sequence transformation using MT models, and it includes a multilingual dataset covering 4 European languages.

**Reasons To Accept:**

The paper is nicely written and includes a thorough discussion of the results and error analysis.

**Reasons To Reject:**

The paper focuses on 4 European languages, which the authors already acknowledge in their limitations, and as such it is difficult to predict how well the proposed methodology would work in other languages.

**Reproducibility:**

4: Could mostly reproduce the results, but there may be some variation because of sample variance or minor variations in their interpretation of the protocol or method.

**Reviewer Confidence:**

3: Pretty sure, but there's a chance I missed something. Although I have a good feel for this area in general, I did not carefully check the paper's details, e.g., the math, experimental design, or novelty.

---

> ### Author Rebuttal · Authors · 2023-08-28
>
> We thank you for your feedback. Since the reason to reject you cite is shared by Reviewer 2, the detailed response to this criticism will be given in our response to Reviewer 2.

---

### Meta-Review · Area_Chair_zmAU · 2023-09-17

**Recommendation:** 4

**Metareview:**

This paper presents a benchmark for evaluating multilingual (four European languages from different families) dialect-to-standard normalization. The main task is to convert a broad phonetic transcription of dialect(s) into orthography of the standard variety. The authors evaluated several seq-2seq and MT systems and different splits and normalizations on existing datasets.

Reasons To Accept:
- The paper compiles a multilingual dataset, and conducts consistent benchmarking across modeling methods, with detailed error analyses. The obtained results are comparable with previous works, while being more unified and observable across language
- Providing different splits of the same dataset to test different scenarios. This is usually rare in many NLP papers but is very important and usually highlights any artificial gains/losses in systems that can go unseen. Even though in this specific work, the splits did not affect the performance of the system, it is still helpful for future system evaluations.
- Appropriateness of the baselines and the different configurations for the systems used for this task.
- The multilingualism of the benchmark.


Reasons To Reject:
- The paper focuses on 4 European languages, which the authors already acknowledge in their limitations, and as such it is difficult to predict how well the proposed methodology would work in other languages.
- The paper introduces a multilingual dataset, compiled entirely of previous works on dialectal research. Regarding this aspect, the paper makes marginal contributions, as the employed corpora are derived completely from previous works, albeit with a few modifications in splitting and preprocessing.

---

### Decision · Program_Chairs · 2023-10-07

**Decision:**

Accept-Findings

**Comment:**

This paper presents a benchmark for evaluating multilingual (four European languages from different families) dialect-to-standard normalization. The main task is to convert a broad phonetic transcription of dialect(s) into orthography of the standard variety. The authors evaluated several seq-2seq and MT systems and different splits and normalizations on existing datasets.

Reasons To Accept:
- The paper compiles a multilingual dataset, and conducts consistent benchmarking across modeling methods, with detailed error analyses. The obtained results are comparable with previous works, while being more unified and observable across language
- Providing different splits of the same dataset to test different scenarios. This is usually rare in many NLP papers but is very important and usually highlights any artificial gains/losses in systems that can go unseen. Even though in this specific work, the splits did not affect the performance of the system, it is still helpful for future system evaluations.
- Appropriateness of the baselines and the different configurations for the systems used for this task.
- The multilingualism of the benchmark.


Reasons To Reject:
- The paper focuses on 4 European languages, which the authors already acknowledge in their limitations, and as such it is difficult to predict how well the proposed methodology would work in other languages.
- The paper introduces a multilingual dataset, compiled entirely of previous works on dialectal research. Regarding this aspect, the paper makes marginal contributions, as the employed corpora are derived completely from previous works, albeit with a few modifications in splitting and preprocessing.